# Development of Novel Small-Molecule Activators of Pyruvate Kinase Muscle Isozyme 2, PKM2, to Reduce Photoreceptor Apoptosis

**DOI:** 10.3390/ph16050705

**Published:** 2023-05-06

**Authors:** Thomas J. Wubben, Sraboni Chaudhury, Brennan T. Watch, Jeanne A. Stuckey, Eric Weh, Roshini Fernando, Moloy Goswami, Mercy Pawar, Jason C. Rech, Cagri G. Besirli

**Affiliations:** 1Department of Ophthalmology and Visual Sciences, Kellogg Eye Center, University of Michigan, Ann Arbor, MI 48105, USA; 2Department of Internal Medicine, Hematology and Oncology, Michigan Center for Therapeutic Innovation, University of Michigan, Ann Arbor, MI 48109, USA; 3Departments of Biological Chemistry and Biophysics, Center for Structural Biology, University of Michigan, Ann Arbor, MI 48109, USA

**Keywords:** pyruvate kinase muscle isozyme 2, photoreceptor, apoptosis, crystallography, neuroprotection

## Abstract

Treatment options are lacking to prevent photoreceptor death and subsequent vision loss. Previously, we demonstrated that reprogramming metabolism via the pharmacologic activation of PKM2 is a novel photoreceptor neuroprotective strategy. However, the features of the tool compound used in those studies, ML-265, preclude its advancement as an intraocular, clinical candidate. This study sought to develop the next generation of small-molecule PKM2 activators, aimed specifically for delivery into the eye. Compounds were developed that replaced the thienopyrrolopyridazinone core of ML-265 and modified the aniline and methyl sulfoxide functional groups. Compound **2** demonstrated that structural changes to the ML-265 scaffold are tolerated from a potency and efficacy standpoint, allow for a similar binding mode to the target, and circumvent apoptosis in models of outer retinal stress. To overcome the low solubility and problematic functional groups of ML-265, compound **2**’s efficacious and versatile core structure for the incorporation of diverse functional groups was then utilized to develop novel PKM2 activators with improved solubility, lack of structural alerts, and retained potency. No other molecules are in the pharmaceutical pipeline for the metabolic reprogramming of photoreceptors. Thus, this study is the first to cultivate the next generation of novel, structurally diverse, small-molecule PKM2 activators for delivery into the eye.

## 1. Introduction

Photoreceptor cell death underlies vision loss in many retinal disorders, such as macular degeneration, inherited retinal diseases, and retinal detachment. Presently, there is a scarcity of treatment options that prevent photoreceptor death, which leaves an urgent unmet need for neuroprotective modalities that improve photoreceptor survival and ultimately prevent vision loss. Recent studies have suggested that the dysregulation of metabolism is a unifying mechanism in photoreceptor cell death and that the metabolic reprogramming of these cells is a potential novel therapeutic strategy to prolong their survival in retinal degenerations [1,2,3,4,5,6,7,8,9].

Photoreceptors have biosynthetic requirements that are among the highest in the body, and to meet these requirements, photoreceptors maintain the ability to perform aerobic glycolysis [10]. This form of glycolysis, which converts glucose to lactate despite the presence of oxygen, allows cells to budget their bioenergetic needs and is critical to photoreceptor function and survival [3,5,11,12]. Pyruvate kinase muscle isoform 2 (PKM2) is a crucial regulatory enzyme of aerobic glycolysis and catalyzes the penultimate step of glycolysis, producing pyruvate and adenosine triphosphate (ATP) from phosphoenolpyruvate (PEP) and adenosine diphosphate (ADP) [13]. In contrast to the isozyme PKM1, which localizes to the inner retina and has constitutively high activity [14], PKM2 is under complex regulation and is principally expressed in photoreceptors in the retina [8,15,16,17,18]. Tetrameric PKM2 is the high catalytic activity state associated with ATP production and catabolism. Nutrient stress, serine, fructose-1,6-bisphosphate, and SAICAR (succinylaminoimidazolecarboxamide ribose-5′-phsophate) promote this state. On the other hand, its non-tetrameric or dimer/monomer state has low catalytic activity; is associated with anabolism; and is favored by high nutrient availability, certain growth factor signaling, and a multitude of post-translational modifications [15].

We previously characterized a murine model that selectively deleted *Pkm2* in photoreceptors [8]. This deletion led to a compensatory upregulation of the PKM1 isoform in photoreceptors as well as increased total PK activity in the retina [8]. Furthermore, under acute outer retinal stress induced by experimental retinal detachment, this conditional knockout mouse exhibited decreased photoreceptor apoptosis and increased survival [8]. From these data, we hypothesized that increasing PKM2 activity pharmacologically would show similar effects on photoreceptor survival. To this end, we published that ML-265, a small-molecule activator of PKM2 [19,20], increased PK activity in vivo in the rat retina and reduced entrance into apoptosis and improved photoreceptor viability in both in vitro and in vivo outer retinal stress models, suggesting that pharmacologically activating PKM2 to reprogram photoreceptor metabolism is a novel therapeutic approach for photoreceptor neuroprotection [9].

While ML-265 is an ideal tool compound to examine this novel neuroprotective strategy, its lack of aqueous solubility and methyl sulfoxide and aniline functional groups preclude its advancement as an ideal intraocular clinical candidate (Figure 1) [21]. The lack of aqueous solubility associated with ML-265 requires formulation with organic solvents (e.g., DMSO) in amounts that prevent safe intraocular delivery due to the known ocular toxicity of these solvents [9]. The aniline moiety of ML-265 is associated with chemically reactive metabolites [22], and the inclusion of racemic functionality, such as the methyl sulfoxide, complicates and increases the cost of the drug development process.

Therefore, in the present study, we sought to develop the next generation of small-molecule PKM2 activators to lay the foundation for future translational and pharmaceutical efforts in the eye. A series of compounds were developed that replaced the thienopyrrolopyridazinone core of ML-265 with a pyridazinoindolone core, which has expanded synthetic accessibility and regiochemical and functional group flexibility, to ultimately replace the aniline and sulfoxide functional groups and incorporate solubilizing functionality. Compound **2** demonstrated that changes to the core scaffold of the tool compound are tolerated from a potency and efficacy standpoint, allow for a similar binding mode to the target, and continue to circumvent apoptosis in in vitro and in vivo models of photoreceptor stress. To overcome the low solubility and problematic functional groups of ML-265, compound **2**’s versatile core structure for the incorporation of diverse functional groups was then utilized to develop novel PKM2 activators with improved solubility, lack of structural alerts, and retained potency and demonstrable activity after intravitreal delivery.

## 2. Results

### 2.1. Synthesis of PKM2 Activators for Intraocular Delivery

Initial medicinal chemistry efforts to transform ML-265 from a pharmacologic tool to an intraocular lead compound sought to replace the thienopyrrolopyridazinone core of ML-265 with a pyridazinoindolone core, which has expanded synthetic accessibility and regiochemical flexibility for solubilizing functional groups, and sought to assess if the aniline and/or methyl sulfoxide functional groups are required for PKM2 activation. The PKM2 activators developed in these initial medicinal chemistry efforts are shown in Table 1 and were generated via Figure 1, Figure 2 and Figure 3 with additional experimental methods and the NMR spectra of each compound presented in the Supporting Information, Appendix A.

Starting from commercially available methyl 6-methyl-1h-indole-2-carboxylate (**6**) and methyl 6-(methylsulfanyl)-1H-indole-2-carboxylate (**7**), the formation of the core tricyclic ring system commenced with aldehyde installation via standard Vilsmeier–Haack reaction conditions [23] with 79% and 83% yields, respectively (Figure 1). Treatment with potassium carbonate and methyliodide (MeI) in dimethylformamide (DMF) resulted in the methylation of the indoles, providing desired products **8** and **9** in good yield (62% and 87% yield, respectively). Subsequent treatment with hydrazine (N_2_H_4_) in ethoxyethanol at an elevated temperature resulted in the formation of **10** and **11** in 58% and 63% yield, respectively.

Compound **1** was generated from 5,7-dimethyl-3,5-dihydro-4H-pyridazino[4,5-b]indol-4-one (**10**) through treatment with benzyl bromide and potassium *tert*-butoxide (KOtBu) in 61% yield (Figure 2). In a similar fashion, compound **5** was generated through alkylation with tert-butyl 3-(bromomethyl)phenylcarbamate and KOtBu to provide **12** in 66% yield. The removal of the *tert*-butylcarbamate protecting group (Boc) under acid-mediated conditions provided compound **5**. Compounds **2**, **3**, and **4** were generated from 5-methyl-7-(methylthio)-3,5-dihydro-4H-pyridazino[4,5-b]indol-4-one (**11**), as shown in Figure 3. The installation of the benzyl functional group was accomplished via treatment with benzyl bromide and KOtBu to produce **13** in 72% yield. The oxidation of thioether **13** with meta-chloroperoxybenzoic acid (mCPBA) provided compound **3** in 53% yield. Compound **14** was generated in 96% yield via the treatment of **11** with *tert*-butyl 3-(bromomethyl)phenylcarbamate and KOtBu. The treatment of **14** with HCl resulted in the loss of the Boc protecting group and the generation of compound **4** in 63% yield. The oxidation of **14** to generate **15** occurred in the presence of mCPBA with a 57% yield, and the subsequent treatment of **15** with acid resulted in the cleavage of the Boc moiety and the generation of compound **2** in 75% yield.

### 2.2. Small Molecules Activate Recombinant PKM2 and Increase PK Activity In Vitro and In Vivo

The potency and efficacy of each compound, **1** through **5**, was assessed with recombinant human PKM2 using an enzyme-coupled assay that uses lactate dehydrogenase and measures the reduction in NADH via absorbance at 340 nm. The tool compound, ML-265, displayed a maximum activation of 292 ± 21% (best-fit value ± std. error) and a half-maximum activating concentration (AC_50_) of 70 ± 17 nM when tested with the human recombinant enzyme (Table 1). These values are similar to our previous study, validating this assay and demonstrating the consistency in our methods [9]. All the compounds developed in this study demonstrated 200% or above maximum activation and nanomolar potency when tested with recombinant human PKM2 (Table 1). To assess the ability of this pyridazinoindolone series of compounds to cross cellular membranes and increase PK activity, we utilized 661W cells, which are immortalized photoreceptor-like cells that express cone markers [24,25,26]. Similar to the results obtained with the recombinant enzyme, this pyridazinoindolone series of compounds demonstrated nanomolar potency when tested in vitro, with compound **2** providing the greatest PK activation (355 ± 26%, Table 1).

The intravitreal injection of medications has revolutionized how numerous retinal diseases are treated and are ever-increasing in ophthalmic clinical practice [27,28]. Therefore, to assess the ability of these small molecules to activate PK in vivo in the rat retina, we utilized this well-accepted drug delivery method. Two microliters of each respective compound at 850 μM or DMSO were intravitreally injected into rat eyes. The vitreous volume of the rat eye was assumed to be 15 μL, giving a final intravitreal concentration of approximately 100 μM for each compound after injection [29,30]. Retinas were harvested four hours after injection, lysed, and assayed for PK activity as previously described [9]. The eyes treated with DMSO served as the control upon which PK activation was based [9]. Since only a single concentration of each compound was tested in vivo, PK activation was reported as the fold change compared to ML-265 (Table 1). Compound **1** did not increase PK activity appreciably above the baseline PK activity noted in those eyes treated with DMSO. Compounds **3**–**5** increased PK activity in the rat retina after intravitreal injection to a greater extent than compound **1,** but it was still less than half of that observed with the tool compound, ML-265. Compound **2**, though, increased PK activity in the rat retina to a similar degree to ML-265, suggesting it may also have a comparable in vivo AC_50_ (Table 1). Correspondingly, compound **2** had the greatest aqueous solubility (99.8 μM) of the pyridazinoindolone series of small-molecule PKM2 activators (Table 1). To confirm that compound **2**, which was a direct comparator to ML-265 with only core scaffold changes, had comparable in vivo potency and activation, 2 μL of increasing concentrations of compound **2**, ML-265, or DMSO was intravitreally injected into rat eyes with the retinas harvested 4 h later, and the PK activity was assayed in the retinal lysate (Figure 2). ML-265 increased PK activity 202 ± 8% with an AC_50_ = 866 ± 176 nM, while compound **2** activated PK up to 183 ± 4% in vivo with an AC_50_ = 299 ± 42 nM.

### 2.3. X-ray Structure of PKM2 in Complex with Compound ***2***

Considering the comparable potency and target activation achieved with compound **2**, the crystal structure of PKM2 in complex with compound **2** was sought to gain insight into its binding mode and assess if changes to the core scaffold resulted in fundamental differences in the target–ligand interactions. The crystal structure of this complex was solved in the I222 space group to 1.84 Å resolution via molecular replacement using a monomer of the activator-bound form of PKM2 (PDB ID: 3ME3). The structure was refined to an *R*-factor of 0.163 and an *R*_free_ equal to 0.190 (Table 2). The asymmetric unit contained one monomer. The monomer within the asymmetric unit contained a single compound **2** molecule, and each tetrameric PKM2 contained two compound **2** molecules with the small-molecule activator spanning the A-A′ interface (Figure 3a), as previously observed in the co-crystal structure of PKM2 and ML-265 [20]. The PKM2:compound **2** monomer also co-purified with an oxalate molecule, which is an analog of PEP, in the active site.

The binding pocket for compound **2** on PKM2 is distinct from that of the active site and the endogenous activator fructose-1,6-bisphosphate, which is in proximity to the C-C’ interface [20]. As a two-fold axis exists at the A-A′ interface, the binding pocket for compound **2** contains equivalent residues from both protomers that make up this dimer, and the small-molecule activator can alternate between orientations based on the two-fold symmetry axis (Figure 3b). In the binding pocket, the pyridazinoindolone core of compound **2** was observed to be sandwiched between two Phe26 side chains with the aniline moiety forming an edge-to-face π-π interaction with a Phe26 side chain. This aniline moiety also forms a direct hydrogen bond to the carbonyl group of Asp354 and a water-mediated hydrogen bond network with the sidechain of Asn318 and carbonyl group of Asp354. The sulfoxide of compound **2** forms a direct hydrogen bond with the main chain nitrogen of Tyr390. Additionally, the carbonyl group of the pyridazinoindolone core is involved in a water-mediated hydrogen bond network with the carbonyl group of Leu353 and the side chain of Asp354 (Figure 3b). Similar key interactions with the protein were previously noted for the tool compound, ML-265 [20].

### 2.4. Compound ***2*** Treatment Reduces Apoptosis and Prevents Cell Death In Vitro

The above experiments demonstrated that replacing the thienopyrrolopyridazinone core of ML-265 with a pyridazinoindolone core is tolerated from a potency and target activation standpoint and also allows for a similar binding mode to the target. To be confident the pyridazinoindolone core could be utilized in future medicinal chemistry efforts, its photoreceptor neuroprotective effect was assessed. We previously showed that ML-265 decreases photoreceptor apoptosis and improves viability in vitro with 661W photoreceptor-like cells and in vivo in an experimental retinal detachment model [9]. As compound **2** is a direct comparator to ML-265 with its pyridazinoindolone core and aniline and methyl sulfoxide functionality and demonstrated similar activation of and binding to PKM2 as ML-265, we first sought to investigate the ability of compound **2** to reduce apoptosis and cell death using a disease-relevant in vitro model. Fas signaling has a critical role in caspase activation and photoreceptor apoptosis after experimental retinal detachment [33,34]. 661W cells die via caspase-mediated mechanisms [24,26,35], and the Fas-ligand (FasL) treatment of these cells leads to the activation of caspases and subsequent death, analogous to the in vivo experimental retinal detachment model [8,36]. Furthermore, similar to photoreceptors, 661W cells express PKM2 (Figure 4a), and these cells have been used to garner information on the role of metabolism in photoreceptor degeneration [37]. As such, this is a suitable in vitro model of photoreceptor stress that can be utilized to assess the consequences of reprogramming metabolism by pharmacologically activating PKM2 [9]. 661W cells were treated with FasL and either compound **2** or vehicle (DMSO). Compound **2** treatment at a concentration near its AC_50_ in cells (Table 1) or at a concentration more than 10-fold greater than its AC_50_ had no significant effect on caspase activity or cell viability, comparable to what we previously reported with ML-265 (Figure 4b–d) [9]. Analogous to previous reports, FasL treatment alone resulted in a statistically significant increase in caspase 8 and 3 and 7 activation and a decrease in cell viability as compared to DMSO or compound **2** treatment alone (Figure 4b–d) [9,36,38]. Compound **2** treatment significantly decreased caspase 8 activation in the presence of FasL at all concentrations tested and significantly decreased caspase 3 and 7 activation at the highest concentration tested (Figure 4b,c). In accordance with this decrease in caspase activation, compound **2** treatment at a concentration of 200 nM significantly improved cell viability as compared to vehicle in the presence of FasL and trended towards improved cell viability at a concentration of 2 μM as well (Figure 4d). Hence, these results demonstrate that compound **2** and its pyridazinoindolone core can circumvent the Fas proapoptotic pathway and promote cell survival in this in vitro model of photoreceptor cell stress, similar to our previous observations with ML-265 [9].

### 2.5. Compound ***2*** Reduces Apoptosis after Experimental Retinal Detachment

Considering compound **2** produced in vivo PK activation that was on par with ML-265 and demonstrated efficacy in circumventing apoptosis in vitro, we next sought to assess the ability of compound **2**, with its changes to the core scaffold, to reduce apoptosis in an experimental model of retinal detachment in rats. Caspase activation, including caspase 8, increases 3 days status post experimental retinal detachment and has been validated to be a marker of photoreceptor apoptosis and the extent of retinal-detachment-induced photoreceptor cell death [8,9,33,36,39]. Additionally, this caspase activation coincides with the peak of the terminal deoxynucleotidyl transferase dUTP nick end labeling (TUNEL) staining of photoreceptors, specifically after experimental retinal detachment in rodents [9,36,40]. So, the detection and quantification of cleaved caspase 8 can be used as a marker of photoreceptor death after experimental retinal detachment and to assess the efficacy of neuroprotective agents [36]. As such, different concentrations of compound **2** or its vehicle, DMSO, were injected at the time of retinal detachment creation in rats, and the retinas were harvested 3 days later to assess caspase 8 activation. Considering their similarities, the concentrations of compound **2** evaluated were chosen to match those previously tested with the tool compound, ML-265 [9]. Compound **2** statistically significantly reduced the percentage of cleaved caspase-8-positive cells in the retina at the highest dose tested as compared to DMSO (Figure 5).

### 2.6. Synthesis of Structurally Diverse PKM2 Activators with Improved Solubility and Retained Potency

Compound 2 demonstrated that structural changes to the ML-265 scaffold are tolerated from a potency and efficacy standpoint (Figure 2), allowed for a similar binding mode to the target (Figure 3), and circumvented apoptosis in in vitro and in vivo models of photoreceptor stress (Figure 4 and Figure 5). It did not solve the issues of solubility or problematic functional groups but rather provided a more versatile template from which to build structurally diverse PKM2 activators to address these critical drug development issues. So, the next phase of medicinal chemistry efforts focused on replacing the aniline and sulfoxide moieties as well as adding solubilizing functional groups. Importantly, the aniline and sulfoxide moieties were shown, via crystallography (Figure 3b), to participate in hydrogen bonding interactions with PKM2 but were determined to not be critical for the activation of PKM2 as the elimination of these functionality in compound **1**, for example, did not demonstrate a decrease in potency (Table 1). Due to the apparent flexibility and functional group tolerance of these positions, we elected to focus on the incorporation of small, basic functionality known to increase the solubility and formulatability with the goal of generating a homogenous aqueous dosing solution. Additionally, with the results in Table 1 showing a correlation between compound solubility and the in vivo activation of the target, the next series of compounds was designed to not only be more soluble than ML-265 and compound 2 but cover a range of solubility levels with the intention of identifying optimal compound properties. The synthetic routes described in Figure 4, Figure 5 and Figure 6 were utilized to generate this next series of PKM2 activators (Table 3) along with the additional experimental methods presented in the Supporting Information, Appendix A.

Commercially available (5-amino-2-fluorophenyl)methanol (**22**) was treated with sodium hydride (NaH) and MeI in dimethylformamide (DMF) to install the desired methyl ether (Figure 4). Subsequent treatment with N-iodosuccinamide (NIS) in acetic acid provided **23** in 68% yield over the two chemical steps. The treatment of **23** with tosyl chloride (TsCl) in the presence of pyridine in dichloromethane resulted in the formation of the sulfonamide (**24**) in 71% yield. Treating compound **24** with tetrakis(triphenylphosphine)palladium (Pd(PPh_3_)_4_), ethyl propiolate, zinc bromide (ZnBr_2_), and diisopropylethylamine (DIPEA) in tetrahydrofuran (THF) at 80 °C allowed for a palladium-mediated cyclization to form the indole, **25**, in 65% yield. The treatment of **25** with potassium hydroxide (KOH) in THF removed the tosylate moiety, and subsequent treatment with MeI in the presence of potassium carbonate (K_2_CO_3_) in DMF provided the desired N-methylindole, **26**, in 92% yield. The treatment of **26** with the Vilsmeier–Haack reagent (generated upon adding phosphorous oxychloride to DMF) installed an aldehyde at the 3-position of the indole in 73% yield [23]. The resulting aldehyde was subjected to hydrazine (N_2_H_4_) in ethoxyethanol at elevated temperatures to provide intermediate **27** in 89% yield.

The alkylation of intermediate **27** with 2-(bromomethyl)-6-methylpyridine, **28**, in the presence of cesium carbonate, Cs_2_CO_3_, in DMF provided pyridine **29** in 77% yield (Figure 5). The treatment of **29** with boron tribromide (BBr_3_) in dichloromethane provided bromide **30**, which was critical for the placement of the structurally diverse functional groups of compounds **16**–**19** (Table 3). Bromide **30** was utilized without purification and treated individually with ethanolamine, morpholine, pyrrolidine, or dimethylamine hydrochloride in the presence of triethylamine in dichloroethane to produce compounds **16**, **17**, **18,** and **19** in 62%, 90%, 59%, and 78% yields, respectively (Figure 5).

The synthesis of compounds **20** and **21** (Table 3) commenced with the alkylation of intermediate **27** with 2-fluorobenzyl bromide, **31**, in the presence of KOtBu in DMF to provide compound **32** in 81% yield (Figure 6). Compound **32** was then treated with boron tribromide in dichloromethane, resulting in bromide **33**. Without purifying it, bromide **33** was subjected to N-Boc-piperazine and triethylamine in dichloroethane to synthesize intermediate **34** with 56% yield. The removal of the tert-butyl carbamate protecting group on intermediate **34** under acid-mediated conditions (4 M HCl in dichloromethane) provided compound **21** in 92% yield. Exposing compound **33** to ethanolamine in the presence of triethylamine in dichloroethane resulted in the formation of compound **20** in 65% yield.

This structurally diverse series of compounds (**16**–**21**) that resulted from the above medicinal chemistry schemes (Figure 4, Figure 5 and Figure 6) demonstrated nanomolar potency and greater than 200% activation of recombinant human PKM2 (Table 3), similar to ML-265 (Table 1). All compounds except compound **17** showed nanomolar potency and greater than 200% PK activation in 661W cells. Importantly, compounds **16**–**21** were determined to have greater aqueous solubility (Table 3) as compared to ML-265 when tested at pH 7.4 (Table 1). The ability to activate the target in vivo was then assessed for compounds **18**–**21**, which span the range of aqueous solubility identified and demonstrate similar potencies to ML-265 in vitro. Two microliters of each respective compound at 8.5 μM (final vitreous concentration of 1 μM) or DMSO was intravitreally injected into rat eyes, and the PK activity was assessed four hours later. All compounds increased PK activity above the baseline observed in eyes treated with DMSO (Figure 6a). Compound **20** increased PK activity in the rat retina to a similar degree to ML-265 after intravitreal injection. Compounds **18** and **19** increased PK activity appreciably in the rat retina after intravitreal injection but approximately 20–30% less than that observed with the tool compound, ML-265. Compound **21** demonstrated approximately a third of the retinal PK activation as that achieved with ML-265. Additionally, compound **19**, chosen as a representative of this more soluble class of PKM2 activators, was selective for PKM2 and did not activate recombinant PKM1 (Figure 6b), retaining a key aspect of the tool compound [20]. Moreover, compound **19** demonstrated similar potency and activation of the target in cellular assays, regardless of whether it was dissolved in an organic solvent (DMSO) or an aqueous-based formulation at neutral pH and 320 mOsm/kg (Figure 6c).

## 3. Discussion

In this study, we synthesized a series of pyridazinoindolone-based PKM2 activators that were capable of activating the recombinant human enzyme with nanomolar AC_50_ and increasing the total PK activity in vitro in 661W cells and in vivo in rat retinas. Compound **2** in this series of activators, with its pyridazinoindolone core and aniline and methyl sulfoxide functionality, circumvented photoreceptor apoptosis in oft-utilized in vitro and in vivo pre-clinical models of outer retinal stress. Building on this versatile, efficacious pyridazinoindolone core structure, a series of novel, structurally diverse PKM2 activators (compounds **16**–**21**) devoid of problematic functional groups and more soluble than the tool compound was then developed with retained potency and activity with the recombinant enzyme and in vitro, as well as with demonstrable activity in vivo after intravitreal injection. These next-generation, small-molecule PKM2 activators with improved solubility, formulate-ability, and retained potency will lay the foundation for future translational and pharmaceutical efforts in photoreceptor neuroprotection.

The potency, selectivity, and commercial availability of ML-265 [19,20] made it an excellent tool compound to demonstrate that PKM2 activation is neuroprotective to photoreceptors [9]. However, its low aqueous solubility, which necessitates organic solvents with known ocular toxicity [9], prevented it from being an ideal intraocular clinical candidate and prompted the exploration of alternative chemotypes to the thienopyrrolopyridazinone heterocycle of ML-265 with the intention of identifying a more versatile and synthetically accessible class of PKM2 activators that can be built upon to circumvent the issues of the tool compound. The pyridazinoindolone core was chosen as it provides expanded synthetic accessibility, functional group compatibility, and greater regiochemical flexibility for solubilizing functional group incorporation.

Compound **2**, which has a pyridazinoindolone core with methyl sulfoxide and aniline functional groups, is a direct comparator to ML-265 and demonstrates that changes to the core heterocycle are tolerated, as compound **2** retained potency with the recombinant enzyme and in 661W cells, improved upon in vivo potency after intravitreal injection in rat eyes (Table 1 and Figure 2), and, importantly, circumvented apoptosis in in vitro and in vivo models of photoreceptor stress (Figure 4 and Figure 5), similar to ML-265 [9]. The comparable activity and pre-clinical efficacy of compound **2** and ML-265 is also consistent with the similar binding mode and molecular interactions observed in the X-ray crystal structures of PKM2 in complex with compound **2** (Figure 3) or ML-265 [20].

Compounds **1** and **3**–**5** demonstrated that the aniline and methyl sulfoxide functional groups are not required for PKM2 activation. Replacing the methyl sulfoxide group with a methyl group while keeping the aniline moiety in place (compound **5**), replacing the aniline with a benzyl group while keeping the methyl sulfoxide in place (compound **3**), or modifying both the methyl sulfoxide with a methyl group and the aniline with a benzyl group (compound **1**) resulted in compounds that demonstrated similar potency and activation of PKM2 (Table 1). These results are similar to the previously published structure–activity studies for ML-265 [21]. Taken together, the methyl sulfoxide and aniline groups, which pose issues in drug development, do not appear to be necessary for PKM2 activation. Furthermore, these structure–activity relationships, in combination with the similar nanomolar potencies observed for compounds **16**–**21** despite their differing functional groups, suggest the importance of the π-π stacking interactions between the core heterocycle and the binding pocket phenylalanine residues (Figure 3b) for PKM2 binding and activation by this series of small-molecule activators.

Removing one or both solubilizing methyl sulfoxide and/or aniline functional groups (compounds **1** and **3**–**5**) resulted in a significant decrease in solubility as well as a significant rightward shift in in vivo potency as compared to compound **2** and ML-265 (Table 1). This finding may suggest that intravitreal injection into the vitreous humor, which is a predominantly aqueous fluid with a water content of at least 98% [41], results in the precipitation or crystallization of the small-molecule activator, preventing significant concentrations of the compound from reaching photoreceptors where PKM2 is predominantly expressed in the retina [8,17]. Even though the aqueous solubility of compound **2** and ML-265 provides for the increased activation of PKM2 in the retina after intravitreal injection as compared to compounds **1** and **3**–**5**, the solubility is still relatively low and requires the inclusion of organic solvents (e.g., DMSO) in amounts that prevent safe intraocular delivery due to the ocular toxicity of these solvents [9]. Hence, we took advantage of the pyridazinoindolone core’s expanded synthetic accessibility and versatility with respect to functional group incorporation and the regiochemical placement of such moieties to next develop novel, structurally diverse PKM2 activators with improved aqueous solubilities and retained potencies (Table 3).

To enhance the solubility, the methyl sulfoxide of compound **2** was replaced with a variety of methylene-linked solubilizing functionalities (Table 3). The morpholine (compound **16**), ethanolamines (compounds **17** and **20**), pyrrolidine (compound **18**), dimethylamine (compound **19**), and piperazine (compound **21**) are all functional groups known to reduce the lipophilicity and enhance the solubility of organic compounds. However, the reduced lipophilicity imparted by these functional groups can attenuate cell permeability. To mitigate any loss in cell permeability via the incorporation of these hydrophilic functionalities, we added an adjacent fluoro group to help mask the polar nature of these functional groups. Gratifyingly, compounds **16**–**21**, with 2- to 7-fold greater aqueous solubilities than ML-265, retained nanomolar potency with the recombinant target enzyme. Additionally, all except compound **17** demonstrated comparable activity levels in vitro in 661W cells to ML-265, and the greater aqueous solubilities of these compounds did not prevent retinal PK activation when delivered intravitreally (Figure 6). The pendent aniline in ML-265 and compounds **2** and **4** was also replaced with 6-methylpyridine (compounds **16**–**19**) and 2-fluorophenyl (compounds **20** and **21**) with no loss of potency, likely due to their ability to participate in similar types of interactions with the protein.

In conclusion, this study is the first to cultivate the next generation of novel, structurally diverse small-molecule PKM2 activators for delivery into the eye. With the improved solubility, lack of structural alerts, and retained in vivo activity of compounds **16**–**21,** organic solvents can be avoided, aniline-derived toxicity can be mitigated, and the drug development process can be streamlined to take metabolic reprogramming from the bench to the bedside for photoreceptor neuroprotection. A scarcity of treatment options currently exists to prevent photoreceptor death and ultimately prevent vision loss. While the ability of these novel, more soluble PKM2 activators to improve photoreceptor survival in pre-clinical models of retinal degeneration needs to be ascertained in future studies, the work herein provides the basis for new therapeutic agents that could impact millions of patients by combating currently untreatable retinal degenerative diseases.

## 4. Materials and Methods

### 4.1. Materials

*Animals.* All animals were treated according to the Association for Research in Vision and Ophthalmology (ARVO) Statement for the Use of Animals in Ophthalmic and Vision Research. The University Committee on Use and Care of Animals of the University of Michigan approved the protocol (Protocol numbers: PRO00011133 and PRO00011135). Adult Brown-Norway rats of both sexes that had been retired from breeding were housed with a 12 h light and 12 h dark cycle and utilized for all in vivo studies.

### 4.2. Cell Culture

The 661W photoreceptor-like cell line was utilized for all in vitro studies and was provided by Dr. Muayyad al-Ubaidi (Department of Cell Biology, University of Oklahoma Health Sciences Center, Oklahoma City, OK, USA) [26]. Dulbecco’s modified Eagle’s medium (DMEM, Thermo Fisher Scientific, Waltham, MA, USA, Cat No. 11995065) supplemented with 10% fetal bovine serum (FBS), 90 units/mL penicillin, 0.09 mg/mL streptomycin, 32 mg/L putrescine, 40 µL/L of β-mercaptoethanol, and 40 µg/L of both hydrocortisone 21-hemisuccinate and progesterone was utilized. Cells were grown at 37 °C in 5% CO_2_ and 95% air.

*Chemicals.* Reagents of analytical grade were purchased from Sigma (St. Louis, MO, USA) or Combi Blocks Inc. (San Diego, CA, USA). ML-265 was obtained through Cayman Chemical (Ann Arbor, MI, USA; CAS 1221186-53-3).

### 4.3. Pyruvate Kinase Activity Enzyme Assay

*Recombinant Enzyme.* An enzyme-coupled assay that measures the depletion of NADH via absorbance at 340 nm and uses lactate dehydrogenase (LDH) was employed to determine the activity of pyruvate kinase (PK) [8,9]. To determine the AC_50_ (concentration of activator necessary to achieve half-maximal activation) of the different small molecules with PKM2, assays were conducted in a 96-well format using 200 μL/well assay volume with 10 nM human recombinant PKM2 (Sigma, St. Louis, MO, USA, Cat No. SAE0021) or PKM1 (Sigma, Waltham, MA, USA, Cat No. SRP0415), differing concentrations of activator, 0.5 mM PEP, 1 mM ADP, 0.2 mM NADH, and 8 U of LDH in a buffer of 50 mM Tris-HCl (pH 7.4), 100 mM KCl, and 5 mM MgCl_2_ as previously described [8,9,17]. A SPECTROstar Omega plate reader (BMG LABTECH Inc., Cary, NC, USA) monitored absorbance at 340 nm. The MARS software suite was utilized to determine initial velocities. Data were normalized to DMSO (dimethyl sulfoxide)-treated enzyme activity, as previously described [9].

*Cell culture.* Next, 661W cells were incubated with either DMSO or different concentrations of small-molecule PKM2 activator for 2 h, lysed and homogenized in RIPA Lysis and Extraction Buffer (Thermo Fisher, Waltham, MA, USA, Cat No. 89900) with protease inhibitors (Complete-Mini, Roche Diagnostics, Indianapolis, IN, USA), and centrifuged at 10,000 rpm for 10 min to remove cellular debris. The supernatant (4–8 μL) was then used to assess the pyruvate kinase activity as described above. The total protein content was determined using the Pierce BCA protein assay kit (Thermo Fisher, Waltham, MA, USA, Cat No. 23225) with the Pierce™ Bovine Serum Albumin Standard Pre-Diluted Set (Thermo Fisher, Waltham, MA, USA, Cat No. 23208) and used to normalize activity as described previously [19].

*Animals.* Intravitreal injections of various small-molecule PKM2 activators or vehicle (DMSO) were performed in Brown-Norway adult rats. The rats were anesthetized with ketamine (90 mg/mL) and xylazine (10 mg/mL) and their pupils dilated with topical phenylephrine (2.5%) and tropicamide (1%) as described previously [9]. A sclerotomy was created with a 25-gauge microvitreoretinal blade (Walcott Rx Products, Ocean View, NJ, USA) just posterior to the limbus. A blunt 35-gauge cannula was inserted through this sclerotomy into the vitreous cavity to deliver 2 μL of different concentrations of small-molecule activator or DMSO. This injection volume produces minimal reflux with adequate reproducibility [29,30]. Fifteen microliters of vitreous volume was assumed in rats to calculate the intravitreal concentrations of the compound after injection [29,30]. As we have previously described, left eyes were treated with small-molecular activator, while right eyes were treated with DMSO, which served as the control upon which pyruvate kinase activation was based [9]. Four hours later, the retina from each rat eye was dissected from the RPE-choroid, homogenized, and lysed in RIPA Lysis and Extraction Buffer with protease inhibitors. The pyruvate kinase (PK) activity assay, as described above, was performed using 4 to 8 microliters of rat retinal lysate, similar to what we and others have previously described [8,9,17]. Analogous to the cell culture assays, the total protein of the lysate was used to normalize pyruvate kinase activity.

### 4.4. Aqueous Solubility of Small Molecule PKM2 Activators

Compounds **1**–**5**, as well as ML-265, underwent aqueous solubility testing at Eurofins Panlabs (St. Charles, MO, USA), a full-service drug discovery contract research organization under contract with the University of Michigan. In brief, each compound was tested at a concentration of 200 μM in PBS, pH 7.4, and room temperature. After a 24-h incubation, the aqueous solubility (μM) was determined using high-performance liquid chromatography (HPLC) with an ultraviolet/visible light (UV/Vis) detector. The area of the principal peak in a calibration standard for each compound (200 μM) in organic solvent (60/40 *v*/*v*% methanol: water) was compared to the area of the corresponding peak in the PBS sample [42]. Compounds **16**–**21** underwent solubility testing at Analiza (Cleveland, OH, USA), a contract research organization. Five hundred microliters of PBS, pH 7.4 was added to each amount of compound provided. The samples were vortexed briefly and then placed on a rotary shaker at 200 RPM for 24 h at ambient temperature. Thereafter, the samples were vacuum filtered through a 0.45-micron membrane, and the filtrates were injected into a nitrogen detector for quantification. The nitrogen detector was calibrated using standards, and the filtrates were quantitated based on the calibration curve.

### 4.5. PKM2 Protein Expression and Purification

A pET28a-LIC expression vector containing PKM2 (residues 1–531) with an N-terminal, non-cleavable His_6_ tag (MGSSHHHHHHSSGLVPRGS) was purchased from Addgene (Watertown, MA, USA, Addgene plasmid # 25360). The plasmid was transformed into Rosetta^2^ cells. The cells were grown in Terrific Broth at 37 °C to an O.D._600_ of 1.0, induced with 0.5 mM Isopropyl β-D-1-thiogalactopyranoside, and the protein was expressed overnight at 18 °C. The cells were pelleted via centrifugation and stored at −80 °C. The frozen cells were thawed and lysed using sonication in buffer containing 10 mM HEPES, pH 7.5, 300 mM KCl, 5 mM imidazole, 5 mM MgCl_2_, 5% glycerol, 2.5 mM TCEP, 0.5% CHAPS, and protease inhibitors. Cellular debris was removed via centrifugation, and cleared supernatant was batch bound to Ni-NTA resin (Qiagen, Germantown, MD, USA) pre-equilibrated in wash buffer (10 mM HEPES, pH 7.5; 300 mM KCl, 5 mM imidazole, 5 mM MgCl_2_, 5% glycerol, and 2.5 mM TCEP) for 1 h at 4 °C. The resin was washed with 15 column volumes of wash buffer and then applied to a column where protein was eluted with 5 column volumes of wash buffer containing 300 mM imidazole. The eluted protein was then incubated for 30 min at 4 °C with 5 molar equivalents of ADP in 5 mM TCEP and 5 mM MgCl_2_, concentrated, and applied to a Supderdex 200 pre-equilibrated with 10 mM HEPES pH 7.5, 150 mM KCl, 5 mM TCEP, 5 mM MgCl_2_, and 5% glycerol. The protein was eluted as a tetramer and was judged to be >95% pure via SDS-PAGE. The purified tetramer was further concentrated to 28 mg/mL and stored at −80 °C.

### 4.6. PKM2 Crystallization and Structure Determination

Prior to crystallization, purified PKM2 was incubated with 10 molar excess of compound **2** overnight at room temperature. The following day, the PKM2:compound **2** complex was set up for crystallization at 20 °C against wells containing 200 mM ammonium sulfate and 20% polyethylene glycol 3350. Crystals appeared within 24 h in drops containing 0.5 μL of protein complex and 0.5 μL of well solution. The crystals continued to grow for 7 days and were harvested and cryoprotected in well solution containing 25% ethylene glycol. Diffraction data were collected at the Advanced Photon Source on the LS-CAT 21-ID-D beamline equipped with an Eiger 9M detector. The data were processed with HKL2000 [43], and the structure was solved via molecular replacement [44] using a monomer of the activator-bound form of PKM2 (PDB ID: 3ME3) as the search model. The structure was solved in space group I222 with one monomer in the asymmetric unit. Coordinates and restraints for compound **2** were created using Grade [45]. The protein complex was iteratively refined and fit to electron density maps using Buster [45] and Coot [46], respectively. The resulting structure showed the activator bound in two orientations; one orientation was supplied by a symmetry mate as compound **2** resides on a symmetry axis. In addition to compound **2**, the protein co-purified with an oxalate in the active site. The N-terminal His-tag along with the first 12 residues of the protein were disordered in the structure. The data collection and refinement statistics for the structure are shown in Table 2.

### 4.7. Western Blot

The 661W cells were plated in 6-well plates overnight in DMEM with 10% FBS. Subsequently, media were replaced with DMEM, no glucose (Thermo Fisher Scientific, Waltham, MA, USA, Cat No. 11966025), supplemented with either 5.5 mM or 25 mM glucose and 10% dialyzed serum. Forty-eight hours later, the cells were harvested and lysed in RIPA Lysis and Extraction Buffer containing protease and phosphatase inhibitors (Cell Signaling, Cat No. 5872). The protein concentration was estimated using the Pierce BCA protein assay kit. Twenty-five micrograms of protein was diluted in Laemmli buffer (Bio-Rad, Hercules, CA, USA, Cat No. 1610747) complemented with β-mercaptoethanol (Millipore-Sigma, Burlington, MA, USA, Cat No. M6250) and run on a 4–20% Mini-PROTEAN^®^ TGX™ Precast Gel (Bio-Rad, Hercules, CA, USA, Cat No. 4561093EDU). The blots were processed and developed as we have previously described [47]. The antibodies that were utilized were as follows: PKM2-specific monoclonal antibody (Proteintech, Rosemont, IL, USA, Cat No. 60268-1-lg) and anti-α-tubulin antibody, mouse monoclonal (Sigma-Aldrich, St. Louis, MO, USA, Cat No. T6199).

### 4.8. Caspase Activity Assay

Caspase 8 and caspase 3/7 activity was measured using luminescent assay kits (Caspase-Glo 8 and 3/7 Assay Systems, Promega, Madison, WI, USA, Cat Nos. G8200 and G8090, respectively). Twenty-four hours prior to treatment, 661W cells were seeded in white-walled 96-well plates at 2500 cells/well. Cells were treated for 2 h with compound **2** or DMSO prior to the addition of 500 ng/mL FasL (Recombinant Mouse Fas Ligand/TNFSF6 Protein, R&D Systems Inc., Minneapolis, MN, USA, Cat No. 6128-SA-025) and 250 ng/mL HA (Hemagglutinin/HA Peptide Antibody, R&D Systems Inc., Minneapolis, MN, USA, Cat No. MAB060) [9]. As previously published, caspase activity was measured 8 h after treatment by incubating cells with substrate for 1 h as per the manufacturer’s instructions. A plate reader luminometer (BMG Labtech, Inc., Cary, NC, USA) assessed luminescence [9,38].

### 4.9. Cell Viability

A luminescent assay kit (RealTime-Glo MT Cell Viability Assay, Promega, Madison, WI, USA, Cat No. G9711) was utilized to assess cell viability. As above, 661W cells were seeded in 96-well plates (Nunc, Rochester, NY, USA) at 2500 cells/well 1 day before treatment [9,38]. The 661w cells were also pre-treated with compound **2** or DMSO for 2 h before adding 500 ng/mL FasL and 250 ng/mL HA. Viability was measured 48 h after the addition of these latter reagents [9]. As with the caspase assays, luminescence was monitored via a plate reader luminometer (BMG Labtech, Inc., Cary, NC, USA).

### 4.10. Experimental Model of Retinal Detachment

Detachments were created in rats as previously outlined [8,33,36,39,48]. In brief, rats were anesthetized with ketamine (100 mg/mL) and xylazine (20 mg/mL), and their pupils were dilated as above. A sclerotomy was created 1–2 mm posterior to the limbus to avoid damaging the lens. Two microliters of varying concentrations of compound **2** (85 μM, 850 μM, or 8.5 mM) or equal-volume DMSO was injected into the vitreous cavity through this sclerotomy at the time of retinal detachment. To create the experimental retinal detachment, a subretinal injector with a 35-gauge beveled cannula was inserted through the sclerotomy and then through a peripheral retinotomy into the subretinal space. Eight microliters of sodium hyaluronate (10 mg/mL) (Abbott Medical Optics, Irvine, CA, USA, Healon OVD) was injected to detach the neurosensory retina from the retinal pigment epithelium (RPE).

### 4.11. Flow Cytometric Assessment of Cleaved Caspase 8 after Experimental Retinal Detachment

PKM2 activator and DMSO-treated detached rat retinas were collected after 3 days to assess the percent of cells positive for cleaved caspase 8 via flow cytometry. The retinas were dissociated in a 0.25% Trypsin solution (Thermo Fisher, Waltham, MA, USA, Cat No. 25200) containing 400 ug of deoxyribose nuclease (Worthington Biochemical, Lakewood, NJ, USA, Cat No. LS002425) and incubated at 37 °C for 20 min with occasional agitation. Dissociated samples were collected via 35 μm nylon mesh, strainer cap flow tubes. The cell suspension was centrifuged at 300× *g* for 5 min at room temperature.

After washing the cells in ice-cold PBS, the cells were fixed with 4% paraformaldehyde for 20 min at room temperature followed by two PBS washes via centrifugation at 300× *g* for 10 min at 4 °C. The cells were then permeabilized using 70% ice-cold methanol at −20 °C overnight. The rinsed cell pellets were incubated with cleaved caspase 8 (Asp387) (D5B2) rabbit monoclonal antibody (Cell Signaling, Danvers, MA, USA, Cat No. 14071) at 1:50 dilution or concentration-matched Rabbit IgG as an isotype control for 1 h at room temperature. The cells were washed with FACS buffer (1% BSA in PBS) and subjected to flow cytometry using an LSR II from BD Biosciences. To exclude events that may represent clumped cells or debris, events were gated based on forward and side scatter. Data were analyzed with FCS Express software (De Novo Software, Ontario, CA, USA).

### 4.12. Statistical Analysis

The results observed in this report are represented as mean ± SEM. Data were analyzed using one-way ANOVA for in vitro and in vivo efficacy data in Prism 9.0 (GraphPad Software, San Diego, CA, USA). The initial velocity data obtained from the steady state kinetic assay described above were analyzed using nonlinear regression curve fitting with the agonist versus response model in Prism 9.0.

## Data Availability

The data generated and analyzed during this study are available from the corresponding author upon reasonable request. The coordinates and structure factors for the X-ray crystallography protein structure reported herein have been deposited in the PDB under accession code 8G2E. The NMR spectra for each compound is presented in Appendix A.

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
