# Peer review of "Development of Novel Small-Molecule Activators of Pyruvate Kinase Muscle Isozyme 2, PKM2, to Reduce Photoreceptor Apoptosis"

_pharmaceuticals, 2023, doi:10.3390/ph16050705_

Round 1

Reviewer 1 Report

There are several potential shortcomings to this study:

1.      The study focuses specifically on developing small molecule PKM2 activators for delivery into the eye. While this is an important area of research, it may not address the larger issue of developing effective treatment options for photoreceptor death and vision loss.

2.      While the study reports promising results for compound 2 and other novel PKM2 activators, it does not provide any information on the safety or efficacy of these compounds in clinical trials. It is important to note that many potential therapies fail to demonstrate efficacy in human trials, so more research is needed to determine whether these compounds will be effective in treating vision loss.

3.      While the study reports that compound 2 has a similar binding mode to the target and circumvents apoptosis in models of outer retinal stress, there is still a risk that these compounds may have unintended effects on other parts of the body. Further research is needed to assess the specificity of these compounds and determine whether they have any off-target effects.

4.      The study only focuses on the development of PKM2 activators for the treatment of vision loss caused by photoreceptor death. It is possible that other approaches, such as gene therapy or stem cell therapy, may be more effective for treating other forms of vision loss.

Moderate editing of English language

Author Response

1. The study focuses specifically on developing small molecule PKM2 activators for delivery into the eye. While this is an important area of research, it may not address the larger issue of developing effective treatment options for photoreceptor death and vision loss.

We appreciate the reviewer’s comment, and we agree that the only way to truly know if this therapeutic strategy will be effective for patients is producing clinical leads that can be taken into human trials. This manuscript details our efforts to begin this important process as there is an urgent unmet need in this area.

2. While the study reports promising results for compound 2 and other novel PKM2 activators, it does not provide any information on the safety or efficacy of these compounds in clinical trials. It is important to note that many potential therapies fail to demonstrate efficacy in human trials, so more research is needed to determine whether these compounds will be effective in treating vision loss.

We agree with the reviewer that further efforts are necessary before these compounds can be brought to the clinic. As noted in the Discussion, studies are ongoing to assess the efficacy and safety of these compounds (lines 571-575).

3. While the study reports that compound 2 has a similar binding mode to the target and circumvents apoptosis in models of outer retinal stress, there is still a risk that these compounds may have unintended effects on other parts of the body. Further research is needed to assess the specificity of these compounds and determine whether they have any off-target effects.

We have demonstrated that this compound class, as represented by Cmpd 19, is specific for the PKM2 isoform (Fig. 6b). That said, the reviewer is correct that any off-target effects and/or safety signals need to be elucidated before these compounds can be taken into the clinic. Those aspects are beyond the scope of this manuscript.

4. The study only focuses on the development of PKM2 activators for the treatment of vision loss caused by photoreceptor death. It is possible that other approaches, such as gene therapy or stem cell therapy, may be more effective for treating other forms of vision loss.

A multitude of therapeutic strategies are being developed by us and others to improve photoreceptor survival and prevent vision loss. Direct comparisons of the different strategies are beyond the scope of this manuscript.

Reviewer 2 Report

The study by Wubben et al. describes that structure modification of ML-265, PKM2 activator, improved solubility, retained in vivo activity and improves its advancement as an intraocular candidate to prevent photoreceptor death and subsequent vision loss. They showed that Compound 2 which was shown to be a comparator to ML-265 was then utilized to develop novel PKM2 activators with retained potency in vitro and in vivo. The authors conclude that improved solubility, overcoming aniline-derived toxicity to a certain limit with these small molecule PKM2 activators can have a therapeutic potential for the treatment of photoreceptor death and subsequent vision loss. This work is interesting,  novel and well written.

Minor comments:

1-    More details can be added about the animals used (Age, Gender, numbers)

2-    FACS plots will be useful to be added (Gating strategy)

Minor changes are required but overall it is well written

Author Response

The study by Wubben et al. describes that structure modification of ML-265, PKM2 activator, improved solubility, retained in vivo activity and improves its advancement as an intraocular candidate to prevent photoreceptor death and subsequent vision loss. They showed that Compound 2 which was shown to be a comparator to ML-265 was then utilized to develop novel PKM2 activators with retained potency in vitro and in vivo. The authors conclude that improved solubility, overcoming aniline-derived toxicity to a certain limit with these small molecule PKM2 activators can have a therapeutic potential for the treatment of photoreceptor death and subsequent vision loss. This work is interesting, novel and well written.

Minor comments:

  • More details can be added about the animals used (Age, Gender, numbers)

In the Materials and Methods section, we have stated the following, “Adult Brown-Norway rats of both sexes that had been retired from breeding were housed on a 12-hour light and 12-hour dark cycle and utilized for all in vivo studies” (lines 582 and 583). The number of animals included in each experiment is noted in the respective figure legend.

  • FACS plots will be useful to be added (Gating strategy)

We have added representative flow cytometry contour plots in Figure 5.